# Tightly coupled inhibitory and excitatory functional networks in the developing primary visual cortex

**Haleigh N Mulholland[1], Bettina Hein[2], Matthias Kaschube[3], Gordon B Smith[1,4]\***

[1]Department of Neuroscience, University of Minnesota, Minneapolis, United States; [2]Center for Theoretical Neuroscience, Zuckerman Institute, Columbia University, New York, United States; [3]Frankfurt Institute for Advanced Studies & Department of Informatics and Mathematics, Goethe University, Frankfurt am Main, Germany; [4]Optical Imaging and Brain Sciences Medical Discovery Team, University of Minnesota, Minneapolis, United States

**Abstract** Intracortical inhibition plays a critical role in shaping activity patterns in the mature cortex. However, little is known about the structure of inhibition in early development prior to the onset of sensory experience, a time when spontaneous activity exhibits long-range correlations predictive of mature functional networks. Here, using calcium imaging of GABAergic neurons in the ferret visual cortex, we show that spontaneous activity in inhibitory neurons is already highly organized into distributed modular networks before visual experience. Inhibitory neurons exhibit spatially modular activity with long-range correlations and precise local organization that is in quantitative agreement with excitatory networks. Furthermore, excitatory and inhibitory networks are strongly co-aligned at both millimeter and cellular scales. These results demonstrate a remarkable degree of organization in inhibitory networks early in the developing cortex, providing support for computational models of self-organizing networks and suggesting a mechanism for the emergence of distributed functional networks during development.

**\*For correspondence:**
gbsmith@umn.edu

**Competing interest:** The authors declare that no competing interests exist.

## Introduction

Inhibition is crucial for shaping neural activity and response properties in the mature cortex. GABAergic interneurons have been implicated in a range of computations, including response gain, stimulus discrimination, and network stabilization (for reviews, see *Isaacson and Scanziani, 2011*; *Denève and Machens, 2016*; *Ferguson and Cardin, 2020*). In the columnar visual cortex, inhibitory neurons actively shape response selectivity (*Wilson et al., 2018*) and are organized into functionally-specific networks with excitatory neurons (*Wilson et al., 2017*). However, relatively little is known about inhibition in the developing cortex prior to the onset of sensory experience. GABAergic inhibition is initially absent in early development before progressively strengthening as the cortex matures (reviewed in *Ben-Ari, 2002*; *Huang et al., 2007*), raising the possibility that inhibition plays only a minor role in shaping early patterns of cortical activity.

Recent work in the ferret visual cortex demonstrated that prior to the onset of visual experience, excitatory activity is already highly structured, showing modular and distributed activity with long-range correlations (*Smith et al., 2018*), reminiscent of the columnar stimulus-evoked activity found in the mature cortex (*Hubel and Wiesel, 1968*; *Blasdel and Salama, 1986*; *Weliky et al., 1996*; *Issa et al., 2000*; *Kara and Boyd, 2009*; *Smith et al., 2015*). These correlated networks are not abolished when silencing feedforward activity and predict future visually-evoked responses (*Smith et al., 2018*), suggesting a key role for early spontaneous activity in the development of mature cortical

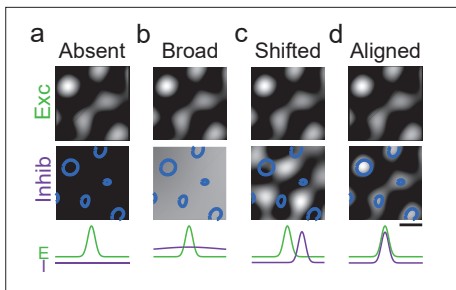

**Figure 1.** Possible modes of inhibition in the developing visual cortex. Schematic showing potential arrangements of inhibitory and excitatory activity patterns in developing cortex. *Top row:* Excitation is known to be highly modular and spatially structured. *Middle row*: Inhibition could be absent (**a**), broadly unstructured (**b**), or modular (**c, d**); if modular, the spatial arrangement of those modules could be shifted with respect to excitatory activity (**c**), or already well aligned as found in the mature cortex (**d**). Scale bar: 1 mm.

networks. However, the structure of inhibitory activity at this early stage of development remains unknown, and several distinct possibilities exist for the state of inhibition in the developing cortex (*Figure 1*): (1) Inhibition could be weak or absent; (2) inhibition may be present but spatially broad and unstructured, operating over a much larger spatial scale than excitation; (3) inhibition might be organized and modular, but poorly aligned with excitation; or (4) the highly organized and co-aligned patterns of excitation and inhibition found in mature animals (*Wilson et al., 2017*) might already be present early in the developing visual cortex. Resolving this question is critical to both constrain models of early network development—many of which rely on structured intracortical inhibition—and to understand the degree to which inhibition contributes to distributed patterns of cortical activity in early development.

Here, we address this by using widefield and two-photon calcium imaging of spontaneous activity to examine the structure of inhibitory networks in the developing ferret primary visual cortex. We first demonstrate in vivo that at early developmental ages where the visual cortex shows modular activity, GABAergic signaling already exerts a strong inhibitory effect. Next, by employing the inhibitory neuron-specific enhancer *mDlx* (*Dimidschstein et al., 2016*) paired with the genetically encoded calcium indicator GCaMP6s, we find that inhibitory neurons exhibit modular patterns of activity that extend over several millimeters in the developing visual cortex. These patterns reveal long-range correlated networks with precise local organization, highly similar in structure to those of excitatory networks. Furthermore, we find that inhibitory and excitatory networks show a remarkable degree of spatial consistency and co-alignment on both global and cellular scales. These findings clearly demonstrate the long-range and fine-scale organization of intracortical inhibition in the

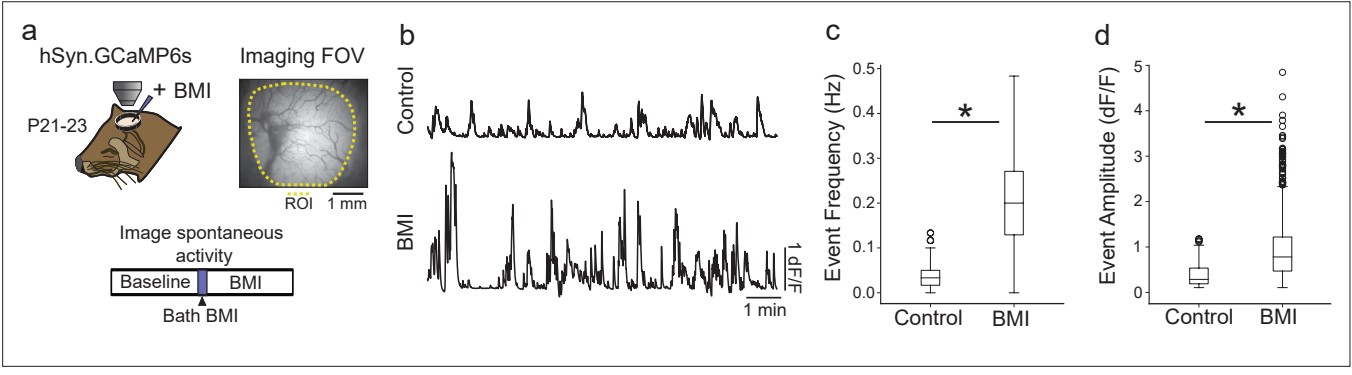

**Figure 2.** GABAergic signaling has net inhibitory effect on cortical activity by P21. (**a**) Experimental schematic; Ferret kits were injected with syn. GCaMP6s at approximately P10, and spontaneous activity was imaged at P21–23 (n=4 animals) both before and after bath application of the GABA antagonist bicuculine (BMI). (*Right*) Field-of-view for widefield epifluorescent calcium imaging. (**b**) Average trace across region of interest (ROI) for baseline spontaneous activity (*top*) and activity after BMI application (bottom) shows clear disinhibition of cortical activity. (**c**) Event frequency after BMI application is significantly higher than baseline (p<0.001; baseline=0.030 [0.016–0.050 ] Hz; BMI=0.200 [0.129–0.271 ] Hz; median, interquartile range [IQR], Wilcoxon rank-sum test). (**d**). Average event amplitude of BMI events are significantly higher than baseline (p<0.001; baseline=0.374 [0.190–0.535] ΔF/F, n (across four animals)=274 events; BMI=0.974 [0.470–1.220] ΔF/F, n=732 events; median, IQR, Wilcoxon rank-sum test).

The online version of this article includes the following source data for figure 2:

**Source data 1.** GABAergic signaling has net inhibitory effect on cortical activity by P21.

developing cortex, and support the ability of large-scale cortical networks to self-organize through precisely correlated local excitatory and inhibitory activity.

## Results

### Spontaneous activity in inhibitory populations is modular and forms large-scale correlated networks prior to eye-opening

Over the course of development, GABAergic signaling switches from exerting depolarizing effects to hyperpolarizing due to the maturation of intracellular chloride concentrations (*Ben-Ari, 2002*). In the ferret, functional GABAergic synapses have been shown to be present as early as postnatal day 20 (P20) (*Dalva, 2010*), but because these experiments utilized whole-cell recordings in cortical slices, it remains unclear whether these synapses exert an inhibitory effect at this age. To address this, we expressed GCaMP6s in excitatory neurons at P21–23 via AAV1.hSyn.GCaMP6s (*Chen et al., 2013 Wilson et al., 2017*) and performed widefield epifluorescent imaging of spontaneous activity in the primary visual cortex (*Smith et al., 2018*) prior to and following direct application of the GABA(A) antagonist bicu-culline methiodide (BMI) to the cortex (*Figure 2a–b*). BMI resulted in a pronounced increase in the frequency (*Figure 2c*; p<0.001; baseline=0.030 [0.016–0.050] Hz; BMI=0.200 [0.129–0.271] Hz, n=4 animals; median, interquartile range [IQR], Wilcoxon rank-sum test) and amplitude of spontaneous events (*Figure 2d*; p<0.001; baseline=0.374 [0.190–0.535] ΔF/F, n [across four animals]=274 events; BMI=0.974 [0.470–1.220] ΔF/F, n=732 events; median, IQR, Wilcoxon rank-sum test), demonstrating that in vivo GABA exerts a strong net inhibitory action at this point of development.

To examine the structure of inhibitory networks in early development, we performed a series of experiments in which GABAergic neurons were specifically labeled via virally mediated expression of GCaMP6s under control of the *mDlx* enhancer, which was previously shown to drive expression specifically in GABAergic cells (*Dimidschstein et al., 2016*; *Wilson et al., 2017*). Widefield imaging of spontaneous activity in the primary visual cortex was performed at postnatal days 23–29 (P23–29, n=7 animals), approximately 7 days before eye-opening (typically between P31 and P34), at a time when modular activity and millimeter-scale networks have been previously shown in excitatory cells (*Smith et al., 2018*; *Figure 3a*).

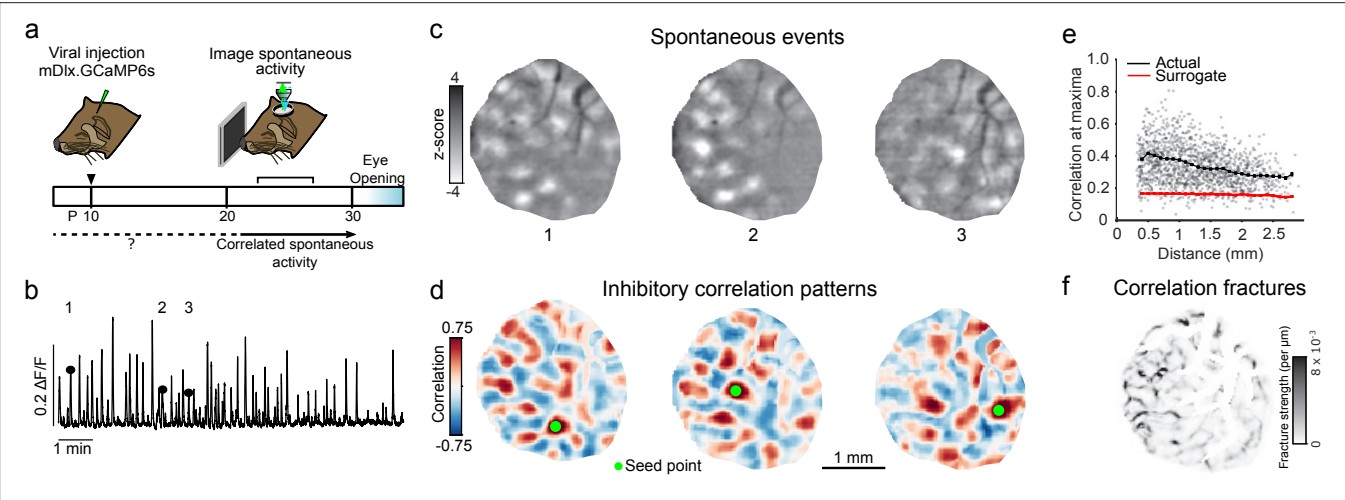

**Figure 3.** Spontaneous inhibitory activity is modular and participates in precisely organized large-scale correlated networks. (**a**) Experimental schematic; ferret kits were injected with inhibitory specific mDlx.GCaMP6s at around P10, and spontaneous activity was imaged prior to eye-opening. (**b**) Example mean trace of inhibitory spontaneous activity across field of view for one representative animal. Numbers correspond to events in (**c**). (**c**) Example single-frame bandpass filtered events. (**d**) Spontaneous correlation patterns for three different seed points. (**e**) Correlation values at maxima as a function of distance from the seed point, as compared to surrogate shuffled correlations for the experiment shown in (**b–d, f**). (**f**) Fractures in correlation patterns reveal regions of rapid transition in global correlation structure.

The online version of this article includes the following figure supplement(s) for figure 3:

**Figure supplement 1.** Spatial extent and variation of spontaneous inhibitory events.

**Figure supplement 2.** Correlation patterns of inhibitory spontaneous activity exhibit pronounced fractures.

Under light anesthesia, inhibitory neuron populations showed frequent spontaneous activity that exhibited a highly modular structure, with spontaneous events containing multiple distinct domains of activity distributed across the imaging field-of-view (FOV) and spanning several millimeters (***Figure 3b–c***, ***Figure 3—figure supplement 1***). To determine if these modular patterns of activity reflect large-scale correlated inhibitory networks, we calculated the pixelwise Pearson's correlation coefficient across all detected spontaneous events. We found that inhibitory neurons indeed exhibit a spatially extended correlation structure, with both strong positively and negatively correlated domains extending several millimeters from the reference seed point (***Figure 3d***). Peak values of positively correlated domains were statistically significant up to 2 mm away from the seed point (***Figure 3e***; $p<0.01$ vs. surrogate data, 6 of 7 individual animals, bootstrap test), demonstrating the large-scale nature of inhibitory networks in the developing visual cortex.

In the mature cortex, orientation preference maps exhibit smooth variation, punctuated by abrupt discontinuities at orientation pinwheels and fractures (***Bonhoeffer and Grinvald, 1991***; ***Ohki et al., 2005***), an organization shared by inhibitory neurons in the mature ferret (***Wilson et al., 2017***). Notably, prior work identified similar discontinuities in the patterns of distributed correlated activity (termed correlation fractures) in excitatory neurons early in development prior to the emergence of orientation maps (***Smith et al., 2018***). To determine if developing inhibitory networks exhibit similar fine-scale organization, we calculated the rate of change in correlation patterns for all pixels within our FOV (***Figure 3—figure supplement 2***). This analysis revealed the presence of spatially discrete fractures over which inhibitory correlation patterns exhibit abrupt changes in structure, demonstrating a precise local organization in the coupling to large-scale inhibitory networks in the developing primary visual cortex (***Figure 3f***).

## Quantitative agreement between inhibitory and excitatory network structure in developing visual cortex

The presence of modular activity patterns, together with long-range correlations exhibiting precise local structure, suggests that inhibitory neurons may already be tightly integrated into functional networks with excitatory cells in the developing cortex. To begin to address this, we undertook a quantitative comparison of excitatory and inhibitory networks assessed through spontaneous activity. We performed widefield imaging in animals expressing either hSyn.GCaMP6s, which is excitatory-specific in ferret (***Wilson et al., 2017***), or the inhibitory-specific mDlx.GCaMP6s. Imaging inhibitory and excitatory activity in separate animals allowed us to compare the statistical properties of spontaneous activity using the same highly sensitive calcium indicator. We first quantified the size of modular active domains in spontaneous events by fitting a two-dimensional Gaussian ellipse to each active region within an event and calculating the full-width at tenth of maximum (FWTM). Domain size was similar between inhibitory and excitatory events (***Figure 4a***; minor axis: I: 412.06 µm [386.73–439.94], n=7 animals; E: 392.68 µm [371.46–397.49], n=7 animals; median [IQR]; p = 0.142, Wilcoxon rank-sum test), and scales together regardless of threshold (full-width half max: I: 226.08 µm [212.19–241.38], E: 215.45 [203.81–218.09], data not shown). Together this supports the idea that the modular domains of local inhibitory activity operate on the same scale as excitatory activity.

Next, we quantitatively examined the correlation patterns for excitatory and inhibitory networks revealed by spontaneous activity. We find that correlations were similarly long-range, with equivalent strength 2 mm away from the seed point (***Figure 4c***; I: r=0.29 (0.25–0.31), E: r=0.24 [0.27–0.31]; p=0.848, Wilcoxon rank-sum test). Likewise, the wavelength estimated from the full angle-averaged correlation function was similar between inhibitory and excitatory networks (***Figure 4d,e***; I: 0.84 mm [0.70–0.96], E: 0.81 mm [0.77–0.901]; p=0.848, Wilcoxon rank-sum test), and approximately similar to the wavelength of orientation columns in the mature cortex (***Bonhoeffer and Grinvald, 1993***; ***White et al., 2001b***; ***Kaschube et al., 2010***). Additionally, correlations in inhibitory networks exhibited a strong local anisotropy, with local correlations around the seed point showing a high degree of eccentricity that was consistent with excitatory correlations (***Figure 4f,g***; I: 0.70 [0.66–0.73], E: 0.69 [0.67–0.71]; p=0.949, Wilcoxon rank-sum test). Finally, we found that the strength of correlation fractures was highly similar between excitatory and inhibitory networks, reflecting a similar degree of precision in fine-scale network organization (***Figure 4h***; I: $1.86*10^{-3}$ (per µm) [$1.49*10^{-3}$ –$2.13*10^{-3}$], E: $1.62*10^{-3}$ (per µm)[$1.35*10^{-3}$–$2.22*10^{-3}$]; p=0.655, Wilcoxon rank-sum test). Taken together, these

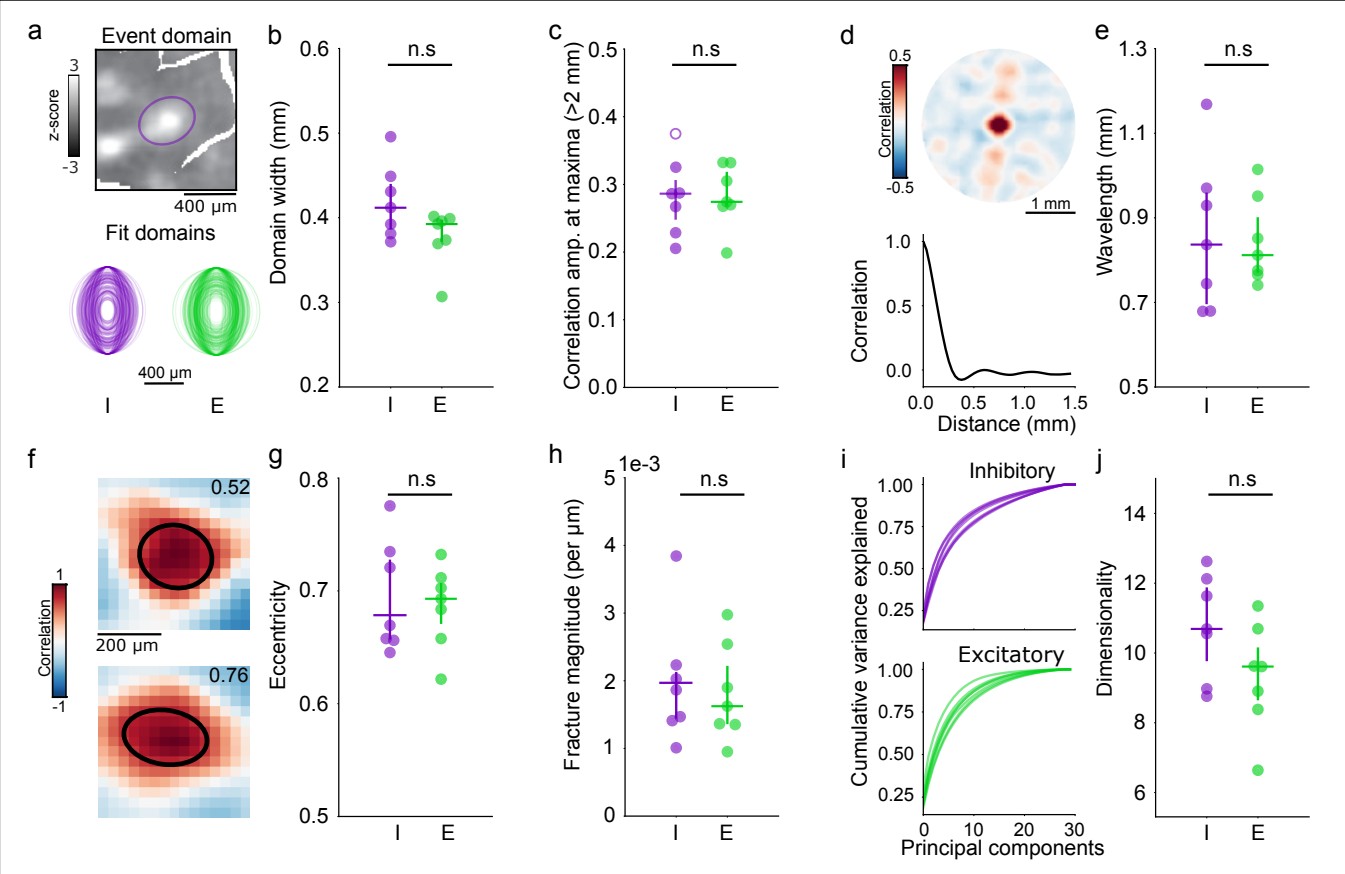

**Figure 4.** Quantitative agreement in structure of developing excitatory and inhibitory networks. (**a, b**) Similar domain size for excitatory and inhibitory events. (**a**) Example event showing FWTM of Gaussian fit of active inhibitory domain (top) and fits of FWTM for 200 randomly selected domains from individual inhibitory (I, purple) and excitatory (E, green) animals, rotated to align major axis (bottom). (**b**) Median domain size for each I and E animal. Circles: individual animals, lines: group median and IQR. (**c**) Correlation strength at 1.8–2.2 mm from seed point. Filled circles individually significant versus surrogate (p<0.01). (**d, e**) Correlation wavelength is similar between I and E networks. (**d**) (top) Correlation pattern averaged over all seed points in single animal with inhibitory GCaMP. (bottom) Correlation as function of distance. (**e**). Wavelength of the correlation pattern for I and E. Circles: individual animals, lines: group median and IQR. (**f–g**) Similar eccentricity of local correlation structure. (**f**) Examples for two seed points from the same inhibitory animal, numbers indicate measured eccentricity. (**g**) Median eccentricity for all seed points. Circles: individual animals, lines: group median and IQR. (**h**) Fracture strength is similar for I and E networks. (**i, j**) Spontaneous activity is moderately low-dimensional in both E and I networks. (**i**) Cumulative explained variance for principal components of spontaneous events (lines indicate individual animals). (**j**). Median dimensionality, calculated from event-matched (n=30 events, n=100 simulations) random subsampling of events. Circles: individual animals, lines: group median and IQR. IQR, interquartile range.

The online version of this article includes the following figure supplement(s) for figure 4:

**Figure supplement 1.** Spatial structure of inhibitory and excitatory correlation patterns from large events are highly similar to those computed from all events.

results demonstrate a strong degree of quantitative similarity in the spatial structure of inhibitory and excitatory networks in the developing visual cortex.

Prior work has shown that in locally heterogeneous network models that recapitulate the structure of developing excitatory networks, long-range correlations exhibiting both local anisotropy and pronounced fractures strongly coincide with spontaneous activity patterns that reside in a low-dimensional subspace (*Smith et al., 2018*). Given the similarities in both local and long-range correlation structure, as well as fracture strength between excitatory and inhibitory networks, we computed the dimensionality of inhibitory spontaneous events and compared them to the dimensionality of event-number-matched events in excitatory neurons. Here, dimensionality is a quantification of the effective number of principal components describing the data (*Abbott et al., 2011*), which can be used to estimate the amount of variation in the observed events. We found that across animals,

inhibitory and excitatory events tended to reside in similarly low-dimensional subspaces (*Figure 4i,j*; I: 10.55 [9.60–11.78], E: 9.44 [8.44–10.01]; median [IQR]; p=0.142, Wilcoxon rank-sum test), further supporting the contribution of inhibition to local heterogeneous networks in the developing visual cortex.

## Precise spatial alignment of inhibitory and excitatory networks across millimeters

These results strongly suggest that excitatory and inhibitory neurons are tightly coupled into the same functional networks early in development. To address this directly, we performed widefield imaging of spontaneous activity of both excitatory and inhibitory neurons within the same animals (P25–26, n=3 animals; *Figure 5—figure supplement 1a*). Animals were injected with AAVs expressing GCaMP6s in inhibitory cells under control of the *mDlx* enhancer (*Dimidschstein et al., 2016*) and jRCaMP1a (*Dana et al., 2016*) in excitatory cells with hSyn (*Wilson et al., 2017*). Spontaneous activity was imaged

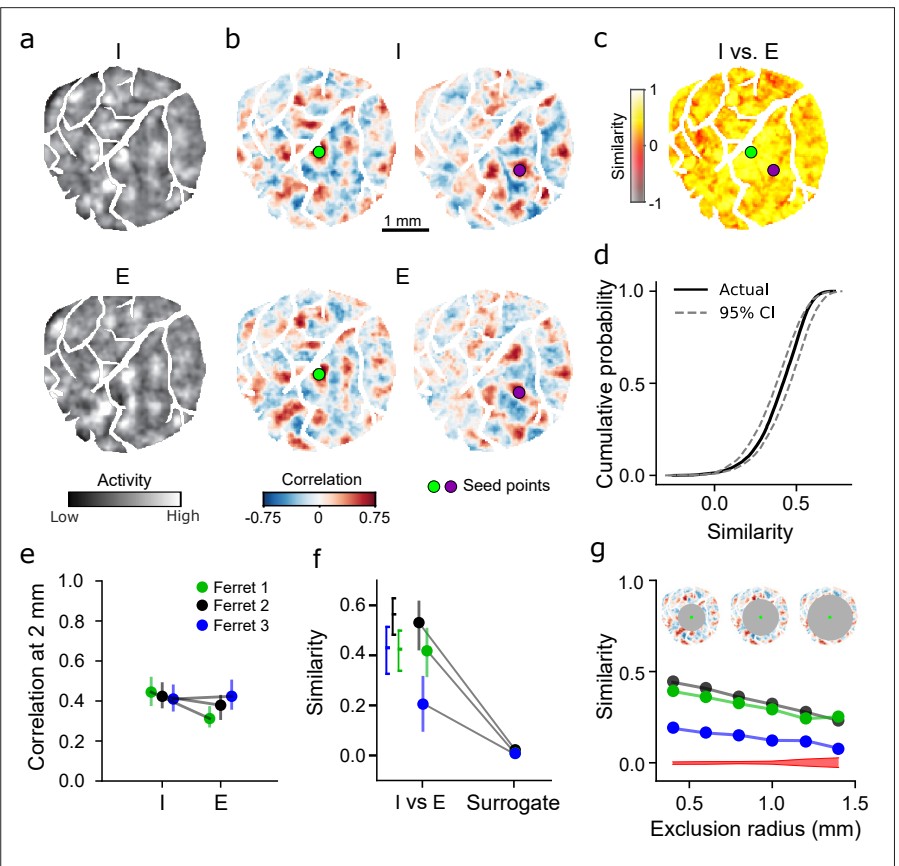

**Figure 5.** Inhibitory and excitatory networks show high degree of similarity within animal. (**a**) Example spontaneous inhibitory (mDlx.GCaMP6s, top) and excitatory (syn.jRCaMP1a, bottom) events recorded from same animal (I color axis: –2 to –2 z-score, E color axis: –1 to 1 z-score). Example events were chosen based on pattern similarity. (**b**) Highly similar correlation patterns in inhibitory and excitatory networks for corresponding seed points. (**c**) Quantification of I versus E correlation similarity for all seed points. Circles indicate seed points illustrated in (**b**). (**d**). Similarity for all seed points in (**c**) falls within 95% confidence intervals (CIs) of bootstrapped I versus I similarity. (**e**) Within animal comparison of correlation strength for E and I (median and IQR within animal). Filled circles individually significant versus surrogate (p<0.01). (**f**) Similarity of I versus E correlations is significantly greater than shuffle (median and IQR within animal). Brackets indicate IQR of bootstrapped I versus I similarity. (**g**) Long-range correlations show significant network similarity. I versus E correlation similarity remains significant versus surrogate (red, 95% CI of surrogate, averaged across animals) for increasingly distant regions (excluding correlations within 0.4–1.4 mm from the seed point).

The online version of this article includes the following figure supplement(s) for figure 5:

**Figure supplement 1.** Similar patterns of spontaneous activity in inhibitory and excitatory events.

using appropriate excitation and emission filters in interleaved blocks of 20 min. We observed similar patterns of modular spontaneous events in inhibitory and excitatory neurons, and individual events with corresponding patterns of activity could frequently be found in both excitatory and inhibitory data sets (*Figure 5a*, *Figure 5—figure supplement 1*).

To compare the structure of excitatory and inhibitory networks, we computed pixelwise correlations separately for all excitatory and inhibitory events for each animal. When selecting the same seed point, the spatial patterns of correlated activity were highly similar across networks (*Figure 5b*), and exhibited equivalent strength extending across several millimeters (*Figure 5e*). To quantify this similarity, we computed second-order correlations between excitatory and inhibitory correlation matrices (see Materials and methods), which revealed high levels of correlation similarity across nearly all seed-points within the imaging window (*Figure 5c*), indicating highly similar large-scale networks. To determine if the level of similarity between networks was statistically significant, we compared it to both the distribution obtained from subsampling inhibitory events (I vs. I, thereby establishing an upper bound given measurement noise and finite event numbers), as well as to surrogate data. We find that excitatory and inhibitory correlation similarity is significantly greater than surrogate ($p < 0.01$ for 3 of 3 individual animals, bootstrap test) and near the level of within inhibitory similarity (*Figure 5d and f*; colors correspond to data from three individual animals). Notably, excitatory and inhibitory networks exhibit strong similarity even in their long-range correlations, as similarity remained significantly greater than surrogate even when considering only correlations over 1.4 mm from the seed point (*Figure 5g*; $p < 0.01$, I vs. E networks vs surrogate, bootstrap test). Taken together, these results demonstrate the presence of overlapping and co-aligned excitatory and inhibitory networks in the developing primary visual cortex.

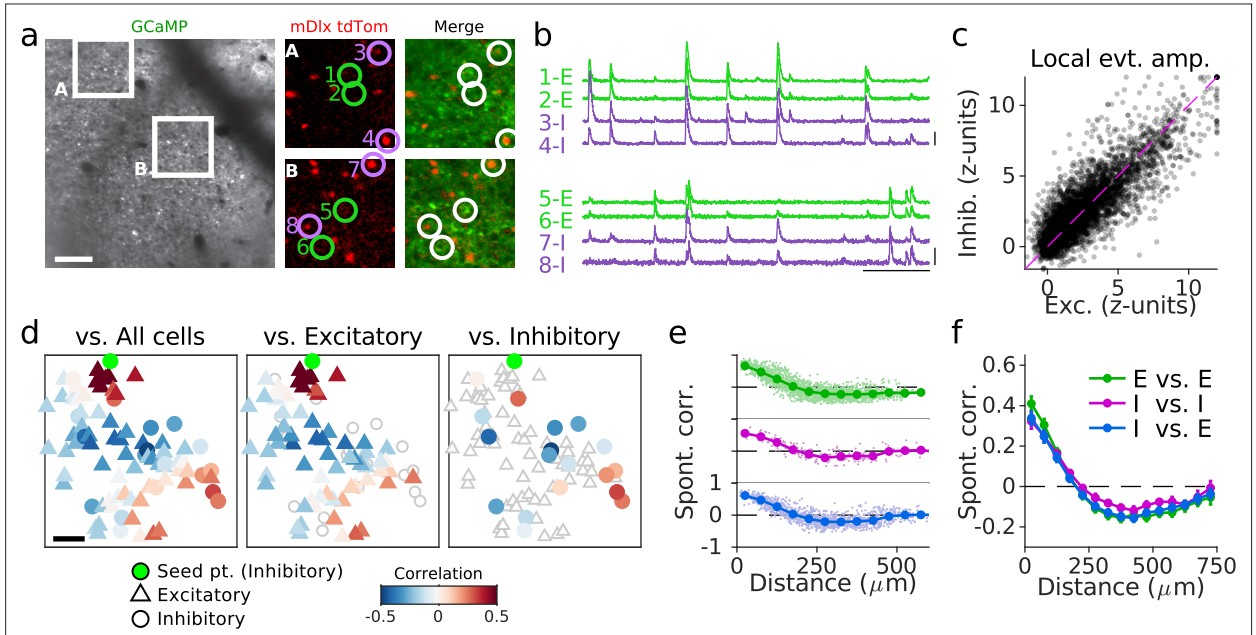

**Figure 6.** Cellular alignment of excitatory and inhibitory networks in developing visual cortex. (**a**) (Left)*:* Example FOV showing GCaMP signal from excitatory and inhibitory neurons. (Middle/Right)*:* Expanded images showing nuclear tdTomato signal specifically in inhibitory neurons (middle) and merged with GCaMP signal (right). (**b**) Traces of spontaneous activity for excitatory (E, green) and inhibitory (I, purple) neurons as indicated in (**a**). (**c**). Event amplitude for local (<75 μm) excitatory and inhibitory responses. For clarity when plotting, 5% of responses are shown for 2186 cells and 1231 events from 21 FOV in five animals, and responses are clipped at 12 z-units. (**d**). Example pairwise correlations between an inhibitory neuron (green) and all cells (left), excitatory cells (middle), and inhibitory cells (right). (**e**). Pairwise correlations as function of distance for an example FOV shown in (**a–d**) for E-E, I-I, and E-I pairs. (**f**) same as (**e**) averaged over 21 FOV from five animals. Scale bars: (**a,** d): 100 μm; (**b**): 30 s, 3 ΔF/F. FOV, field-of-view.

The online version of this article includes the following figure supplement(s) for figure 6:

**Figure supplement 1.** tdTomato expressed from AAV1.mDlx.GCaMP6s.P2A.NLS-tdTomato specifically labels GABAergic neurons.

**Figure supplement 2.** Spontaneous activity patterns are spatially modular and aligned across excitatory and inhibitory neurons.

**Figure supplement 3.** Examples of co-aligned correlation structure between excitatory and inhibitory cells.

## Excitatory and inhibitory networks are tightly integrated at cellular scale

To assess whether the large-scale alignment of excitatory and inhibitory networks observed above extends to the cellular level, we performed simultaneous two-photon imaging of excitatory and inhibitory neurons (P23–26). We co-injected AAVs expressing hSyn.GCaMP6s (excitatory) and mDlx. GCaMP6s.P2A.NLS-tdTomato (inhibitory), allowing us to distinguish inhibitory neurons in vivo through tdTomato expression (*Figure 6a*, *Figure 6—figure supplement 1*). We observed highly modular spontaneous activity in both excitatory and inhibitory cells, with local populations showing tightly coordinated patterns of activity across cell types (*Figure 6b*, *Figure 6—figure supplement 2*).

To quantify the relative balance of excitatory and inhibitory activity within local populations, we computed the average response of all cells of a given cell type within a 75-µm region around each neuron for all spontaneous events. We found the amplitude of events within these local populations was highly correlated between excitatory and inhibitory cells (r=0.86, 2186 cells and 1231 events from 21 FOV in five animals; *Figure 6c*), suggesting locally balanced activity. Both inhibitory and excitatory neurons exhibited spatially organized pairwise correlations across spontaneous events, with distributed patches of positively correlated cells located hundreds of microns away. Notably, the spatial structure of pairwise correlations for a given neuron (whether inhibitory or excitatory) with either inhibitory or excitatory cells was well aligned (p<0.001 vs. shuffle), irrespective of cell type (*Figure 6d*, *Figure 6—figure supplement 3*). Finally, examining correlation strength as a function of distance and cell type demonstrates that correlated excitatory and inhibitory networks exhibit a similar spatial scale at the cellular level (*Figure 6e and f*). Taken together, these results show that excitatory and inhibitory neurons are precisely organized in the developing visual cortex into the same spatially structured and locally correlated functional networks.

## Discussion

By applying the inhibitory interneuron-specific expression of fluorescent calcium sensors to the developing visual cortex, we were able to directly assess the structure of inhibitory cortical networks and their integration with excitatory networks. We show that prior to eye-opening and the onset of reliable stimulus-evoked responses (*Chapman et al., 1996*), intracortical inhibition is both present and highly organized, with inhibitory networks already displaying patterns of modular and correlated spontaneous activity that span several millimeters. Inhibitory networks exhibit quantitatively similar structures to excitatory networks, which together show precise alignment at both local and global scales in the patterns of correlated spontaneous activity. Taken together, these findings demonstrate that the presence of tightly coupled excitatory and inhibitory functional networks in the developing visual cortex.

In the mature ferret, both inhibitory and excitatory neurons are organized into a columnar map of orientation preference, with orientation-specific subnetworks of functionally coupled excitatory and inhibitory neurons present not only within iso-orientation domains but also near orientation pinwheels (*Wilson et al., 2017*). This organization stands in contrast to the non-specific local pooling of nearby excitatory activity by inhibitory neurons found in the 'salt-and-pepper' rodent cortex (*Kerlin et al., 2010*; *Bock et al., 2011*; *Hofer et al., 2011*; *Packer and Yuste, 2011*; *Runyan and Sur, 2013*; *Scholl et al., 2015*), raising the possibility that the functionally specific organization of inhibitory cells in the mature ferret emerges from an initially non-specific state over the course of development. However, our finding of sharp correlation fractures in inhibitory networks (*Figure 3f*, *Figure 3—figure supplement 2*) is inconsistent with non-specific local pooling and argues strongly against this possibility. Rather, our results indicate that in the developing visual cortex, inhibitory neurons are already tightly integrated into functionally specific networks. Furthermore, given that the patterns of correlated activity undergo extensive refinement in the week prior to eye-opening (*Smith et al., 2018*), these results also suggest that excitatory and inhibitory networks likely refine in parallel over development to produce the tightly coupled organization found in the mature cortex (*Wilson et al., 2017*).

One potential caveat to these results is that genetically encoded calcium sensors can exhibit non-linearities in their response to neural activity that may vary by cell type, baseline firing rate, and sensor, possibly obscuring subtle differences in the structure of excitatory or inhibitory networks. Although such differences could potentially impact the measured size or magnitude of domains or

underestimate the spatial extent of some of the activity patterns, differences in sensor function cannot account for the presence of modular inhibitory activity or the spacing of excitatory and inhibitory domains. Our data clearly show modular patterns of inhibitory activity that extend throughout the full imaging area (*Figure 3—figure supplement 1c-d*), which show clearly detectable troughs between active regions (*Figure 3c*, *Figure 3—figure supplement 1a*), results which also argue against a saturation of the calcium signal. In addition, excitatory events imaged with jRCaMP1a (*Figure 5*) have spatially similar features to those imaged with GCaMP6s (*Figure 5*, *Smith et al., 2018*), indicating that variations in sensitivity or non-linearities across sensors are unlikely to obscure differences in the precise alignment of excitatory and inhibitory correlation patterns.

The *mDlx* enhancer used in this study has been shown to drive expression across several subtypes of inhibitory cells (*Dimidschstein et al., 2016*), which have been ascribed distinct roles within cortical circuits (see (*Hattori et al., 2017*; *Wood et al., 2017*) for reviews). However, in the mature ferret, parvalbumin (PV), somatostatin (SOM) and non-PV, non-SOM GABAergic neurons were all found to be equivalently integrated into specific and spatially organized functional networks (*Wilson et al., 2017*). Although the low expression of PV in the weeks prior to eye-opening (*Gao et al., 2000*) makes it difficult to separate inhibitory neurons by subtype at these ages, our results suggest that a similarly organized and integrated structure to that in the mature ferret may already exist in the developing cortex. Future work, potentially leveraging inhibitory subtype-specific viral approaches (*Mehta et al., 2019*; *Vormstein-Schneider et al., 2020*), will be required to directly test the functional roles of specific GABAergic subtypes in developing cortical networks.

Large-scale distributed functional networks are a hallmark of mature cortical organization and are exemplified by the columnar arrangement of orientation preference in the visual cortex of primates and carnivores (*Blasdel and Salama, 1986*; *Bonhoeffer and Grinvald, 1991*; *Ohki et al., 2005*). Here, co-tuned columns preferentially interconnected by specific long-range horizontal projections (*Gilbert and Wiesel, 1989*; *Malach et al., 1993*; *Bosking et al., 1997*) which emerge over the course of development (*Ruthazer and Stryker, 1996*; *Borrell and Callaway, 2002*). The presence of modular activity and long range-correlations in the developing cortex prior to the emergence of these horizontal connections, coupled with the finding that correlations persist in the absence of feedforward input (*Smith et al., 2018*), has been taken to indicate that local intracortical circuits are sufficient to generate these patterns of activity. Models of developing cortical networks can rely on local connectivity to self-organize, thereby producing modular patterns of activity (*von der Malsburg, 1973*; *Swindale, 1997*; *Miller, 1994*; *Barrow et al., 1996*). If these local connections are sufficiently heterogeneous, the resulting patterns of activity reside in a low-dimensional subspace and produce long-range correlations without requiring long-range horizontal connections (*Smith et al., 2018*). A strong prediction of such models is the presence of tightly integrated and co-aligned excitatory and inhibitory networks (*Ermentrout and Cowan, 1979*; *Wilson and Cowan, 1973*), in agreement with our experimental observations in the developing cortex. Thus, our results suggest that short-range intracortical interactions between tightly coupled excitatory and inhibitory circuits give rise to large-scale distributed networks during early development, which may serve as a seed for future functional organization in the cortex.

# Materials and methods

## Animals
All experimental procedures were approved by the University of Minnesota Institutional Animal Care and Use Committee and were performed in accordance with guidelines from the US National Institutes of Health. We obtained 20 male and female ferret kits from Marshall Farms and housed them with jills on a 16 h light/8 h dark cycle. No statistical methods were used to predetermine sample sizes, but our sample sizes are similar to those reported in previous publications.

## Viral injection
Viral injections were performed as previously described (*Smith and Fitzpatrick, 2016*) and were consistent with prior work (*Smith et al., 2018*). Injections targeted the primary visual cortex, defined as thalamo-recepient visual cortex. In the ferret, primary visual cortex includes both areas 17 and 18, which both receive direct thalamic innervation and exhibit columnar mapping of orientation (*White*

*et al., 1999*; *White et al., 2001a*). Briefly, we expressed GCaMP6s (*Chen et al., 2013*) in inhibitory interneurons by microinjecting AAV1.mDLx.GCaMP6s.P2A.NLS.tdTomato (University of Minnesota Viral Vector and Cloning Core), based on the mDlx inhibitory specific enhancer (*Dimidschstein et al., 2016*), into layer 2/3 of the primary visual cortex at P10–15 approximately 10–15 days before imaging experiments. To image excitatory and inhibitory activity, animals used for widefield experiments were injected with a 1:1 ratio of AAV1.mDlx.GCaMP6s.P2A.NLS.tdTomato and AAV1.Syn.NES.jRCaMP1a. WPRE.SV40, and animals used for two-photon experiments were injected with a 1:1 ratio of AAV1. mDLx.GCaMP6s.P2A.NLS.tdTomato and AAV1.Syn.GCaMP6s.WPRE.SV40.

Anesthesia was induced with isoflurane (3.5–4%) and maintained with isoflurane (1–1.5%). Buprenorphine (0.01 mg/kg) and either atropine (0.2 mg/kg) or glycopyrrolate (0.01 mg/kg) were administered, as well as 1:1 lidocaine/bupivacaine at the site of incision. Animal temperature was maintained at approximately 37°C with a water pump heat therapy pad (Adroit Medical HTP-1500, Parkland Scientific). Animals were also mechanically ventilated and both heart rate and end-tidal $CO_2$ were monitored throughout the surgery. Using aseptic surgical technique, skin and muscle overlying visual cortex were retracted, and a small burr hole was made with a handheld drill (Fordom Electric Co.). Approximately 1 µl of virus contained in a pulled-glass pipette was pressure injected into the cortex at two depths (~200 µm and 400 µm below the surface) over 20 min using a Nanoject-II (World Precision Instruments). The craniotomy was filled with 2% agarose and sealed with a thin sterile plastic film to prevent dural adhesion.

## Cranial window surgery

On the day of experimental imaging, ferrets were anesthetized with 3–4% isoflurane. Atropine was administered as in virus injection procedure. Animals were placed on a feedback-controlled heating pad to maintain an internal temperature of 37–38°C. Animals were intubated and ventilated, and isoflurane was delivered between 1% and 2% throughout the surgical procedure to maintain a surgical plane of anesthesia. An intraparietal catheter was placed to deliver fluids. EKG, end-tidal $CO_2$, and internal temperature were continuously monitored during the procedure and subsequent imaging session. The scalp was retracted and a custom titanium headplate adhered to the skull using C&B Metabond (Parkell). A 6–7 mm craniotomy was performed at the viral injection site and the dura retracted to reveal the cortex. One 4 mm cover glass (round, #1.5 thickness, Electron Microscopy Sciences) was adhered to the bottom of a custom titanium insert and placed onto the brain to gently compress the underlying cortex and dampen biological motion during imaging. The cranial window was hermetically sealed using a stainless-steel retaining ring (5/16-in. internal retaining ring, McMaster-Carr). Upon completion of the surgical procedure, isoflurane was gradually reduced (0.6–0.9%) and then vecuronium bromide (0.4 mg/kg/hr) mixed in an LRS 5% Dextrose solution was delivered IP to reduce motion and prevent spontaneous respiration.

## Widefield epifluorescence and two-photon imaging

Widefield epifluorescence imaging was performed with an sCMOS camera (Zyla 5.5, Andor; Prime BSI express, Teledyne) controlled by µManager (*Edelstein et al., 2010*). Images were acquired at 15 Hz with 4×4 binning to yield 640×540 pixels (Zyla) or 2×2 binning and additional offline 2×2 binning to yield 512×512 pixels (Prime BSI). Two-photon imaging was performed with a commercial microscope (Neurolabware) driven by an Insight X3 laser (Spectra Physics). Imaging was performed at 920 nm (GCaMP) and 1040 nm (tdTomato), and fluorescence was collected on separate PMTs using a 562-nm dichroic mirror and 510/84 nm (GCaMP) and 607/70 nm (tdTomato) emission filters (Semrock). Images were collected at 796×512 pixels at 30 Hz.

Spontaneous activity was captured in 10-min imaging sessions, with the animal sitting in a darkened room facing an LCD monitor displaying a black screen. All imaging of spontaneous activity was done in young animals (P23–29) prior to eye-opening (~P31 to P35).

## Immunostaining and imaging

To confirm that tdTomato expression from AAV1.mDLx.GCaMP6s.P2A.NLS.tdTomato was specific to inhibitory neurons as expected from previous work (*Wilson et al., 2017*), we performed immunostaining in a subset of animals. Following imaging, animals were euthanized and transcardially perfused with 0.9% heparinized saline and 4% paraformaldehyde. The brains were extracted, post-fixed

overnight in 4% paraformaldehyde, and stored in 0.1 M phosphate buffer solution. Using a vibratome, brains were tangentially sectioned along the surface of the imaging window (50 μm steps). Slices were stained for GAD67 using Mouse anti- GAD67 (1:1000, Sigma-Aldrich, MAB5406) and Alexa 405 donkey anti-mouse (1:500, Abcam, AB175659) as described (*Wilson et al., 2017*). Imaging was performed on a confocal microscope (Nikon C2).

## Bicuculline application

To test whether GABAergic signaling exerted net inhibitory effects at the ages examined in this study, we bath applied the GABA(A) antagonist BMI directly to the cortex at P21–23 in animals expressing AAV1.Syn.GCaMP6s.WPRE.SV40. To apply BMI, the cannula covering the cranial window was removed and the exposed cortex was gently flushed with ACSF. After collecting 10 min of baseline spontaneous activity, 10 μM BMI in ACSF was bath applied to the cortex, and spontaneous activity was imaged for an additional 10 min.

## Data analysis

### Signal extraction for widefield epifluorescence imaging

Image series were motion corrected using rigid alignment and a region of interest (ROI) was manually drawn around the cortical region of GCaMP expression. Additionally, a ROI mask was manually drawn around blood vessels to remove vessel artifacts. The baseline fluorescence (F0) for each pixel was obtained by applying a rank-order filter to the raw fluorescence trace with a rank 70 samples (for excitatory data) or 190 samples (for inhibitory data) and a time window of 30 s (451 samples). The rank and time window were chosen such that the baseline faithfully followed the slow trend of the fluorescence activity. The baseline-corrected spontaneous activity was calculated as (F–F0)/F0=Δ F/F0.

### Event detection

Detection of spontaneously active events was performed essentially as described (*Smith et al., 2018*). Briefly, we first determined active pixels on each frame using a pixelwise threshold set to 3 s.d. above each pixel's mean value across time. Active pixels not part of a contiguous active region of at least 0.01 mm$^2$ were considered 'inactive' for the purpose of event detection. Active frames were taken as frames with a spatially extended pattern of activity (>50% of pixels were active). Our results were not sensitive to area threshold, as removing the area threshold requirement and including all frames with activity greater than 3 s.d resulted in correlation patterns with highly similar spatial structure for both inhibitory and excitatory networks. Furthermore, the strength of correlations 2 mm from the seed point for inhibitory and excitatory networks were equivalent for networks calculated from large events and all events (*Figure 4—figure supplement 1*). Consecutive active frames were combined into a single event starting with the first high-activity frame and then either ending with the last high-activity frame or, if present, an activity frame defining a local minimum in the fluorescence activity. To assess the spatial pattern of an event, we extracted the maximally active frame for each event, defined as the frame with the highest activity averaged across the ROI. While correlation patterns have been shown to be stable with as low as 10 events (*Smith et al., 2018*), datasets with fewer than 30 detected events were excluded from this study.

Due to differences in signal-to-noise ratio in experiments using jRCaMP1a, we used a pixelwise threshold of 2 s.d to determine active pixels, and frames were >30% of the pixels were active were considered active frames.

### Spontaneous correlation patterns

Spontaneous correlation patters were calculated as previously described (*Smith et al., 2018*). Briefly, we applied a Gaussian spatial band-pass filter (sigma$_{low}$=26 μm and sigma$_{high}$=195 μm) to the maximally active frame in each event and downsampled it to 160×135 (128×128, Prime BSI) pixels. The resulting patterns, named spontaneous patterns A in the following, were used to compute the spontaneous correlation patterns as the pairwise Pearson's correlation between all locations x within the ROI and the seed point s:

$$C\left(s, x\right) = \frac{1}{N-1} \sum_{i=1}^{N} \frac{\left(A_i\left(s\right) - \langle A_i\left(s\right)\rangle\right)\left(A_i\left(x\right) - \langle A_i\left(x\right)\rangle\right)}{\sigma_s \sigma_x}$$

Here the brackets ⟨ ⟩ denote the average over all events and $\sigma_x$ denotes the standard deviation of A over all N events i at location x.

## Spontaneous fractures

Spontaneous fractures were computed as previously described (*Smith et al., 2018*). Fracture strength was defined as the rate by which the correlation pattern changes when moving the seed point location over adjacent pixels. We defined fracture magnitude as the difference in fracture strength averaged over the fracture lines, and its average in regions >130 μm apart from the nearest fracture line. Fracture lines were identified by first applying a spatial median filter with a window size of 78 μm to remove outliers. We then applied histogram normalization contrast enhancement using contrast-limited adaptive histogram equalization (CLAHE, clip limit=20, size of neighborhood 260×260 μm² using 'createCLAHE' in Python OpenCV), and a spatial high-pass filter (Gaussian filter, s.d. sigma-$_{high}$=390 μm). The resulting values were binarized (threshold=0), and the resulting two-dimensional binary array eroded and then dilated (twice) to remove single noncontiguous pixels (Python, Multidimenisonal image processing library [scipy.ndimage]). We skeletonized this binary array to obtain the fracture lines (Python, Scikit-Image).

## Event domain size

To estimate the size of active domains in spontaneous events, we first identified domains by taking the local maxima of each event after bandpass filtering. The local neighborhood of each domain (600 μm radius from maxima) was then fit with a two-dimensional Gaussian using nonlinear least squares:

$$a = \frac{cos^2\theta}{2\sigma_x^2} + \frac{sin^2\theta}{2\sigma_y^2}$$

$$b = \frac{-sin2\theta}{4\sigma_x^2} + \frac{sin2\theta}{4\sigma_y^2}$$

$$c = \frac{sin^2\theta}{2\sigma_x^2} + \frac{cos^2\theta}{2\sigma_y^2}$$

$$G = y_0 + He^{\left(-\left(a(x-x_1)^2 + 2b(x-x_1)(y-y_1) + c(y-y_1)^2\right)\right)}$$

where H is the amplitude of the Gaussian, $x_1$ ($y_1$) is the x (y) center, $\sigma_x$ ($\sigma_y$) is the standard deviation of the x (y) component, $\theta$ is the angular rotation of the Gaussian, and $y_0$ is the offset. Domain size was calculated as the FWTM of the minor axis of fitted Gaussian:

$$FWTM_{minor} = \sigma_{minor}2\sqrt{2\ln(10)}$$

## Strength of long-distance correlations

To determine the strength of correlations, we first identified local maxima (minimum separation between maxima: 800 μm) in the correlation pattern for each seed point. To assess the statistical significance of long-range correlations ~2 mm from the seed point, we compared the median correlation strength for maxima located 1.8–2.2 mm away against a distribution obtained from 100 surrogate correlation patterns. Surrogate correlation patterns control for correlations that arise from finite sampling by eliminating most of the spatial relationship between patterns (*Smith et al., 2018*). Surrogate correlation patterns were generated from spontaneous events that were randomly rotated (rotation angle drawn from a uniform distribution between 0° and 360° with a step size of 10°), translated (shifts drawn from a uniform distribution between ±450 μm in increments of 26 μm, independently for x and y directions) and reflected (with probability 0.5, independently at the x and y axes at the center of the ROI).

## Correlation pattern wavelength

To find the wavelength of the correlation patterns, we first centered and averaged the local neighborhood (1500 μm radius) across all seed points. We then averaged over the angle to obtain the average correlation as a function of distance from the seed point and used spline interpolation to fit the data. The wavelength of the resulting spline interpolation was estimated as the distance to the first local maxima after 0.

## Eccentricity of local correlation structure

To describe the shape of the local correlation pattern around a seed point, we fit an ellipse (least-square fit) with orientation $\Phi$, major axis $\varsigma_1$ and minor axis $\varsigma_2$ to the contour line at correlation = 0.7 around the seed point. The eccentricity $\varepsilon$ of the ellipse is defined as:

$$\varepsilon = \frac{\sqrt{\varsigma_1^2 - \varsigma_2^2}}{\varsigma_1}$$

Where $\varepsilon$ =0 is a circle, and increasing values indicating greater elongation along the ellipse.

## Dimensionality of spontaneous activity

We estimated the dimensionality $d_{eff}$ of the subspace spanned by spontaneous activity patterns by *Abbott et al., 2011*:

$$d_{eff} = \frac{\left(\sum_{i=1}^{N} \lambda_i\right)^2}{\sum_{i=1}^{N} \lambda_i^2}$$

where $\lambda_i$ are the eigenvalues of the covariance matrix for the N pixels within the ROI. As the value of the dimensionality is sensitive to differences in detected event number, to estimate the distribution of the dimensionality for each animal we calculated the dimensionality of randomly sub-sampled events (n=30 events, matched across animals, 100 simulations) and took the median of the distribution.

## Comparison of inhibitory and excitatory correlation similarity

To compare the similarity between inhibitory and excitatory correlation patterns within the same animal, we computed the second-order correlation between patterns. For each seed point, we calculated the second order Pearson's correlation between corresponding correlation patterns, while excluding pixels within a 400-μm exclusion radius around the seed point to prevent local correlations from inflating the similarity between the two networks. To get an estimate the upper bound of similarity within inhibitory networks, given a finite sampling size, we randomly split the detected inhibitory events into two groups and separately computed correlations and the second-order correlations between the halves (n simulations=100.) To determine if the observed networks are more similar than chance, we calculated the similarity between the excitatory network and a surrogate inhibitory network (surrogate events calculated as above, with 100 simulations and surrogate similarity calculated as median and IQR across simulations). To determine if spontaneous correlations far from the seed point also maintain high degrees of similarity, we systematically increased the size of the exclusion radii, calculating similarity only using data far from the seed points. Exclusion radius size ranged from 400 to 1400 μm, in 200 μm steps.

## Two-photon event detection and cellular correlations

Two-photon images were corrected for rigid in plane motion via a 2D cross-correlation. Cellular ROIs were drawn using custom software (Cell Magic Wand; *Wilson et al., 2017*) in ImageJ and imported into Matlab via MIJ (*Sage and Prodanov, 2012*, http://bigwww.epfl.ch/sage/soft/mij/). Cellular ROIs were then manually categorized as either being inhibitory or excitatory based on tdTomato expression. Fluorescence was averaged over all pixels in the ROI and traces were neuropil subtracted, where $F_{neuropil}$ was taken as the average fluorescence within 30 μm excluding all cellular ROIs and $\alpha$=0.6:

$$F_{cell} = F_{raw} - \alpha F_{neuropil}$$

Activity was taken as ΔF/F0, where F0 was the baseline fluorescence obtained by applying a 60-s median filter, followed by a first-order Butterworth high-pass filter with a cutoff time of 60 s. To compute spontaneous correlations, we first identified frames containing spontaneous events, which were defined as frames in which >10% of imaged neurons exhibited activity >2 s.d. above their mean. Cellular activity on all event frames was then z-scored using the mean and s.d. of each frame, and pairwise Pearson's correlations were computed across all neurons over all active frames.

The similarity of cellular correlation patterns for inhibitory and excitatory cells was computed by first creating a spatially matched set of excitatory and inhibitory neurons. For each inhibitory cell in the FOV, a spatially matched excitatory cell was identified by finding the closest excitatory cell (within 50 μm).

Then, for every cell in the FOV (excitatory and inhibitory) pairwise correlations were calculated with either inhibitory cells or the spatially matched excitatory cells. The similarity of these correlation patterns was taken as the second-order Pearson's correlation as above, while excluding cells <200 μm from the seed neuron. Statistical significance was computed by comparing measured similarity values against a shuffled distribution obtained by shuffling the pairwise correlations before computing similarity (100 shuffles).

To compare the amplitude of events within a local area, activity traces for each cell were smoothed with a three-sample median filter and z-scored across all frames. We then computed the average amplitude of all inhibitory (or excitatory) cells within a 75-μm radius for the maximally active frame of each event.

## Quantification and statistical analysis

Nonparametric tests were used for statistical testing throughout the study. Bootstrapping was used to determine null distributions when indicated. Center and spread values are reported as median and IQR, unless otherwise noted. Statistical analyses were performed in MATLAB and Python, and significance was defined as $p < 0.05$.

## Acknowledgements

The authors wish to thank Drishti Lall, Casey Xamonthiene, Hailey Glewwe, and Matt Paruzynski for histology and surgical assistance, and members of the Smith and Kaschube labs for helpful discussions. The authors were supported by NIH R01EY030893-01 (GBS), T32 MH115886 (HM), BMBF 01GQ2002 (MK), NSF 1707398 (BH), Gatsby Charitable Foundation GAT3708 (BH), Whitehall Foundation 2018-05-57 (GBS), as well as support from NIH grants P41 EB027061 and P30 NS076408. All viral vectors used in this study were generated by the University of Minnesota Viral Vector and Cloning Core (Minneapolis, MN). This work was supported by the resources and staff at the University of Minnesota University Imaging Centers (RRID: SCR_020997).

## Additional information

### Funding

| Funder | Grant reference number | Author |
| --- | --- | --- |
| National Eye Institute | NIH R01EY030893-01 | Gordon Smith |
| National Institute of Mental Health | T32 MH115886 | Haleigh N Mulholland |
| Bundesministerium für Bildung und Forschung | BMBF 01GQ2002 | Matthias Kaschube |
| National Science Foundation | NSF 1707398 | Bettina Hein |
| Gatsby Charitable Foundation | GAT3708 | Bettina Hein |
| Whitehall Foundation | 2018-05-57 | Gordon Smith |

The funders had no role in study design, data collection and interpretation, or the decision to submit the work for publication.

### Author contributions

Haleigh N Mulholland, Data curation, Formal analysis, Investigation, Methodology, Software, Validation, Visualization, Writing - original draft, Writing – review and editing; Bettina Hein, Methodology, Resources, Software, Writing – review and editing; Matthias Kaschube, Funding acquisition, Methodology, Resources, Software, Supervision, Writing – review and editing; Gordon B Smith, Conceptualization, Data curation, Formal analysis, Funding acquisition, Investigation, Methodology, Project administration, Resources, Software, Supervision, Validation, Visualization, Writing – review and editing

## Author ORCIDs
Haleigh N Mulholland http://orcid.org/0000-0002-8958-8706
Bettina Hein http://orcid.org/0000-0002-5317-1976
Matthias Kaschube http://orcid.org/0000-0002-5145-7487
Gordon B Smith http://orcid.org/0000-0002-1107-3884

## Ethics
All experimental procedures were approved by the University of Minnesota Institutional Animal Care and Use Committee (IACUC) (protocol 1708-35068A) and were performed in strict accordance with guidelines from the US National Institutes of Health. All surgery was performed under isoflurane anesthesia, and every effort was made to minimize suffering.

## Decision letter and Author response
Decision letter https://doi.org/10.7554/eLife.72456.sa1
Author response https://doi.org/10.7554/eLife.72456.sa2

## Additional files

### Supplementary files
• Transparent reporting form

### Data availability
All data generated or analyzed during this study are included in the manuscript and supporting file. The data and code (Python, MATLAB) used in this work are accessible at https://github.com/mulho042/SpontaneousInhib.git, (copy archived at https://archive.softwareheritage.org/swh:1:rev:a17e6056a04d5ce6574d077c5553819b4843f922).

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
