## [Decision Letter]

**Decision letter after peer review:**

Thank you for submitting your article "Tightly-coupled inhibitory and excitatory functional networks in the early visual cortex" for consideration by *eLife*. Your article has been reviewed by 3 peer reviewers, and the evaluation has been overseen by a Reviewing Editor and Tirin Moore as the Senior Editor. The following individual involved in review of your submission has agreed to reveal their identity: Geoffrey J Goodhill (Reviewer #1).

Essential revisions:

All reviewers were in agreement that this study represents significant advancement in our understanding of inhibitory cortical networks during this crucial time in development. There were several suggestions by the reviewers that you are encouraged to implement to clarify the results and address particular confounding factors (particularly of those raised by Reviewer #2). These do not require new experiments but should be addressed in the revised text.

*Reviewer #2 (Recommendations for the authors):*

– My biggest concern is the potential of confounds introduced by using calcium imaging to measure responses of cell types with different firing rates. It would improve the work to more properly discuss these issues, and possibly try to estimate how much of an impact they could have. Is it possible that the extent of the excitatory events are underestimated because their lower firing rates put them below the threshold of measurable GCaMP activity? How different are the non-linearities between the 2 sensors? Finally, how much do the chosen event detection parameters (such as the requirement that more than 50% of pixels have to be active for an event to be counted) influence the results?

– It would also be good to give a sense of how homogenous virus expression was throughout each imaging region, and to rule out that the scale of events is limited by the expression pattern.

– Along the same lines: How many different patterns of spontaneous activity are usually detected? Figure 3 shows 3 evens that all look highly similar (with subtle differences in the most active regions, but not the spatial layout); supplementary figure 1 shows 2 patterns that appear to be the opposite of each other. The correlation matrix in supplementary figure 2 also seems to suggest that there are no more than 4 patterns that occur. The low dimensionality of the subspace supports this conclusion as well, but it would be good to quantify the number of observable patterns more directly (and to compare them against the pattern of expression).

*Reviewer #3 (Recommendations for the authors):*

The paper is quite polished and in my opinion there is little to change.

The methods were generally quite clear and used fundamental methods, but the description of the calculation of spontaneous fractures was opaque to me. It would help to unpack this a bit more and to describe which software tools were used (is CLAHE in ImageJ? Matlab?). Are "eroded" and "diluted" precise actions in a software tool?

---

## [Author Response]

Essential revisions:All reviewers were in agreement that this study represents significant advancement in our understanding of inhibitory cortical networks during this crucial time in development. There were several suggestions by the reviewers that you are encouraged to implement to clarify the results and address particular confounding factors (particularly of those raised by Reviewer #2). These do not require new experiments but should be addressed in the revised text.

We thank the reviewers and editors for their evaluation of our work, and for their suggestions to improve the clarity of our manuscript. In our revised submission, we have addressed all of the concerns raised. We believe the manuscript has been greatly strengthened by these changes, and hope you will find it suitable for acceptance and publication.

A detailed response to each of the points raised in review follows below.

Reviewer #2 (Recommendations for the authors):– My biggest concern is the potential of confounds introduced by using calcium imaging to measure responses of cell types with different firing rates. It would improve the work to more properly discuss these issues, and possibly try to estimate how much of an impact they could have. Is it possible that the extent of the excitatory events are underestimated because their lower firing rates put them below the threshold of measurable GCaMP activity? How different are the non-linearities between the 2 sensors? Finally, how much do the chosen event detection parameters (such as the requirement that more than 50% of pixels have to be active for an event to be counted) influence the results?

Although it is possible that differences in the transformation of spiking activity to GCaMP fluorescence between cell types obscure subtle differences in the structure of excitatory and inhibitory networks, it is unlikely that this is a major factor in our results. If present, such differences could potentially impact the measured size or amplitude of domains. However, differences in calcium reporter behavior would not affect the spatial location and spacing of excitatory and inhibitory domains, and therefore is unlikely to affect in the precise co-alignment of excitatory and inhibitory correlation patterns.

Our results show modular inhibitory activity that extends throughout the full imaging area (Figure 3, Figure 3—figure supplement 1) with clearly visible troughs between active domains, arguing against sensor saturation. Additionally, although GCaMP6s is far more sensitive than jRCaMP1a with a greater signal-to-noise ratio, both sensors reveal clearly modular patterns of spontaneous activity. Furthermore, the activity patterns imaged in excitatory cells with jRCaMP1a are highly similar in structure to those imaged previously in excitatory cells with GCaMP6s (Smith *et al.,* 2018), arguing against jRCaMP1a inducing a prominent distortion of excitatory activity in these experiments. In addition, when comparing inhibitory GCaMP6s and excitatory GCaMP6s in separate animals (Figure 4) we found a high degree of quantitative agreement in the spatial structure of spontaneous events. Together, these points suggest that it is unlikely that variations in sensitivity or non-linearity between GCaMP6s and jRCaMP1a obscure major differences in the spatial patterns of activity in excitatory and inhibitory neurons.

We have expanded our discussion in the main text to explicitly address this point.

To address the concern that our results may be impacted by the choice of event detection parameters, we have now included new data showing that our results are robust to these choices (Figure 4—figure supplement 1). This analysis demonstrates including many smaller and weaker events does not impact the strength or spatial structure of correlations.

– It would also be good to give a sense of how homogenous virus expression was throughout each imaging region, and to rule out that the scale of events is limited by the expression pattern.

As in prior work (Smith et al., 2015, Smith et al., 2018), the viral injection approach used here results in strong expression typically over a >3mm diameter area, with gradual fall-off beyond this distance. In most cases, this results in viral expression filling the entire field of view of the camera. Any imaged areas with poor expression were excluded from the area used for analysis. Notably, any variations in viral expression are at a much larger spatial scale (several mm) than the size of the individual domains (typically half a mm or less) in the modular patterns of spontaneous activity we observe, making it highly unlikely that variations in viral expression within the imaged area contribute to the observed structure of activity.

To further address this point, we have now included a new supplemental figure (Figure 1—figure supplement 1) showing randomly chosen example events which reveal spontaneous activity spanning the full FOV. We quantify this by calculating the fraction of spontaneous events in which a given pixel was active (defined as greater than 3 s.d above its mean activity across time, Figure 1—figure supplement 1b-e). All pixels within our ROI participated in at least 10% of all events, demonstrating that we are able to detect spontaneous activity throughout our imaging area. Together, this indicates that the expression pattern is not a limiting factor in determining the spatial scale of spontaneous activity.

– Along the same lines: How many different patterns of spontaneous activity are usually detected? Figure 3 shows 3 evens that all look highly similar (with subtle differences in the most active regions, but not the spatial layout); supplementary figure 1 shows 2 patterns that appear to be the opposite of each other. The correlation matrix in supplementary figure 2 also seems to suggest that there are no more than 4 patterns that occur. The low dimensionality of the subspace supports this conclusion as well, but it would be good to quantify the number of observable patterns more directly (and to compare them against the pattern of expression).

The reviewer raises an important point, as the spontaneous activity patters in the cortex are indeed low dimensional, but are also more varied than being comprised of only 4 patterns. We recognize that our initial choice of example events in Figure 3c—in which two events (left and middle) were intentionally selected to show patterns with similar spatial layout that appear minutes apart from one another—may have given an incorrect impression of the variability across events.

To better represent the actual variation in the events that we observe, we have changed the third example in Figure 3c to a pattern with a more distinct spatial arrangement from the first two. We have also now included a new Figure 1—figure supplement 1, showing 36 randomly selected examples of inhibitory events from the same animal that contribute to the correlation patterns in Figure 3d. These examples show the wide variety of events observed in spontaneous activity, with modular patterns that span the full area of our imaging field. For quantification, we estimated the variety of patterns by computing their dimensionality (Abbott et al., 2011), which can be summarized as the approximate number of linear combinations of principal component patterns used to describe the variation across all events. The low dimensional subspace that the events reside in indicates that there is indeed a limited repertoire of possible patterns, but that the majority of the variance can be explained by approximately 10-12 patterns, rather than 4. This low dimensional subspace is consistent with previous work observing excitatory spontaneous activity in the developing visual cortex (Smith et al., 2018).

Consistent with this interpretation, the correlations in Supplementary Figure 2 (now Figure 5—figure supplement 1), while showing some positive clusters, are relatively low, indicating a degree of inconsistency even amongst events with similar spatial motifs, and therefore using clustering methods to definitively quantify the number of observed patterns is challenging. Our methods to quantify dimensionality are a more unbiased approach to estimate the variation in the data.

To address these questions, we have made revisions to the Results section of the main text to better clarify and explain our interpretation of dimensionality of spontaneous activity.

Reviewer #3 (Recommendations for the authors):The paper is quite polished and in my opinion there is little to change.

We thank the reviewer for their positive assessment of our manuscript, and appreciate the feedback. We have addressed their suggestions below.

The methods were generally quite clear and used fundamental methods, but the description of the calculation of spontaneous fractures was opaque to me. It would help to unpack this a bit more and to describe which software tools were used (is CLAHE in ImageJ? Matlab?). Are "eroded" and "diluted" precise actions in a software tool?

Fractures were calculated using custom Python code built with open-source libraries. CLAHE was calculated using OpenCv’s function “createCLAHE”, and binary erosion and dilution (these operations are precisely defined in the theoretical framework ‘mathematical morphology’) was calculated using SciPy’s Multidimensional Image Processing library (scipy.ndimage). We have updated the description of fracture magnitude calculation in the methods, and the actual code used to produce fractures can be found in the github repository associated with this paper: (https://github.com/mulho042/SpontaneousInhib.git, see: plotEvents_correlationPatterns.py)